# Changes in the Glutinous Rice Grain and Physicochemical Properties of Its Starch upon Moderate Treatment with Pulsed Electric Field

**DOI:** 10.3390/foods10020395

**Published:** 2021-02-11

**Authors:** Shuang Qiu, Alireza Abbaspourrad, Olga I. Padilla-Zakour

**Affiliations:** Department of Food Science, Cornell University, Stocking Hall, Ithaca, NY 14853, USA; sq59@cornell.edu (S.Q.); Alireza@cornell.edu (A.A.)

**Keywords:** glutinous rice grain, pulsed electric field, porosity, microstructure, gelatinization

## Abstract

Pulsed electric field (PEF) processing is an emerging non-thermal technology that shows potential to improve food quality and to maintain stability. Glutinous rice is composed mainly of amylopectin and has low amylose content. This study investigated the effect of PEF treatment at 3 kV/cm field strength for 50 to 300 pulses on whole, water-soaked glutinous rice grains. Micro-pores were created at the surface of PEF treated rice grains, increasing grain porosity from 7.3% to 9.8%. Peak viscosity of PEF treated rice flour decreased, and breakdown, final and setback viscosities increased as the number of PEF treating pulses increased, indicating that the swelling degree of rice starch was promoted after PEF treatment. Lower values of gelatinization enthalpy and lower crystalline degree of PEF treated glutinous rice flour were also observed. Fourier transform infrared (FTIR) and nuclear magnetic resonance (NMR) studies confirmed the secondary structure changes in rice protein and partial gelatinization of rice starch after PEF treatment.

## 1. Introduction

Pulsed electric field (PEF) treatment is a non-thermal technology that generates short pulses of high electric voltage field strength on sample materials that are placed between two electrodes [1]. During the last two decades, PEF has been considered as one of the emerging food technologies and has received considerable attention due to its potential to improve food quality. PEF is able to alter the textural and physicochemical properties of food products by inducing molecular and configurational modifications on the biomacromolecules of raw materials [2]. PEF can also protect food attributes as a result of less degradation of nutritional components and better preservation of sensory properties during processing [2].

Liquid food materials are electric conductive as they contain ions and electrolytes, thus liquid foods are desired media for PEF treatment. The reported applications of PEF have been mainly in liquid foods, such as fruit juice [3], dairy products [1], egg liquids [4], and alcoholic beverages [1]. In a variety of liquid food processes, PEF treatment facilitated mass transfer, improved the diffusion of cellular liquids which improved food dehydration [5] and promoted extraction efficiency of functional compounds from food plants [6]. PEF was also utilized in fruit and vegetable processing to deactivate microorganisms [1], to reduce pesticide residues [7], and to inactivate enzymes [8].

Recent studies have shown that PEF could change the properties of solid food materials as well. The impacts of PEF on the biomacromolecules of food substance will result in irreversible changes on the solid food products after processing. The electroporation induced by PEF could lead to the breakage of lignin-cellulose bonds, which resulted in a softer texture of white asparagus [9]. In addition, PEF-induced electroporation could increase the moisture content of potato and softened potato chunks without any modifications on the fresh appearance. Potato cubes after PEF treatment absorbed less oil and showed less browning/bleaching effects upon frying than control samples [10].

Rice is one of the most widely consumed staple foods for the world’s human population, especially in Asian countries. There are many rice varieties showing diverse textural properties after cooking, and they are utilized as various culinary materials depending on the acceptance of consumers from different areas of the world. Glutinous rice (*Oryza sativa*), also called sticky rice or waxy rice, is composed mainly of amylopectin and has very low amylose content. Compared to non-glutinous rice, the cooked glutinous rice grains are sticky and can provide the product with a flexible and characteristic texture as the amylopectin mainly contributes to the viscous properties of gelatinized starch. Cooked glutinous rice grains exhibit a slower rate of retrogradation as the branched amylopectin structure showed much less tendency of retrogradation than amylose. Glutinous rice is the main ingredient to make many traditional Asian foods and snacks such as glutinous rice balls (*Tang-yuan*), rice dumplings (*Zong-zi*), rice crackers (*Mi-gao*) and glutinous rice cakes (*Mochi*) [11]. Those glutinous rice products prepared with adequately cooked glutinous rice grains or glutinous rice flour show a flexible and characteristic texture as the cooked glutinous rice is sticky and adhesive [12]. The cooked glutinous rice also has a slower rate of retrogradation and hardening compared to cooked normal rice [12,13].

Recent studies have shown that electric field based techniques (PEF, Ohmic heating, moderate electric field and induced electric field) have potential to modify the properties and functions of starch in a clean, efficient and homogeneous manner [14]. According to Han et al., [15,16], 50 kV/cm of PEF treatment could generate partial gelatinization and reduction of degree of crystallinity of potato, maize and cassava starch. PEF treatment significantly promoted the acetylated substitution degree of cassava and potato starches during chemical modification. High intensity of PEF treatment (40–50 kV/cm) also induced crystallinity and structural changes in waxy rice starch, and affected its digestibility [17]. However, to the best of our knowledge, little is known about the application of PEF on whole rice grains which is a more common and less expensive food ingredient than isolated starch. The physicochemical and microstructural changes of raw glutinous rice grains induced by PEF treatment could lead to significant modifications on the final glutinous rice products. According to our ongoing studies, glutinous rice products prepared using PEF treated rice grains possessed softer texture, smoother mouthfeel, inhibited retrogradation properties and extended shelf life compared to control glutinous rice products (under investigation). Therefore, PFE treatment can be utilized as a novel processing method to potentially develop glutinous rice products with improved textural properties and stability. It is much more difficult and energy-consuming to generate high PEF field strength (30–50 kV/cm) than treating materials with longer time at low and achievable PEF field strength for industrial application. Generally, smaller electrode gap and less chamber volume were applied to achieve high electric field strength which greatly restricted the efficiency of the PEF treatment. Thus, longer treating time or more pulses at moderate PEF field strength (less than 5 kV/cm) offer promising opportunities to conduct further research.

The objectives of the current work were to investigate the effects of PEF treatment on glutinous rice grains. In this study, native polished glutinous rice grains were treated with 50–300 PEF pulses at the field strength of 3kV/cm. The physicochemical properties and gelatinization behaviors of glutinous rice grains prepared with or without PEF treatment were investigated. Results from the present study will enhance the understanding of the role of PEF on glutinous rice grains (waxy starch/amylopectin), and also advance the utilization of PEF technology for industrial cereals processing.

## 2. Materials and Methods

### 2.1. Material

Commercial polished glutinous rice grains (from Thailand) were obtained from a local supermarket in Ithaca, NY. The moisture, crude protein, total starch, crude fat and ash content were determined by Dairy One Forage Laboratory (Dairy One Inc., Ithaca, NY, USA). The amylose content was quantified using the colorimetric method described by McGrance et al. [18].

### 2.2. Pulsed Electric Field (PEF) Treatment of Rice

Dry glutinous rice grains were soaked in adequate amounts of distilled water at room temperature for 2 h before PEF treatment. Afterwards, soaked grains were drained and washed twice using distilled water to remove damaged or small particles.

Lab-scale PEF Pilot^TM^ unit (Elea GmbH, Quakenbrück, Germany) was used to process the pre-treated rice grains. Soaked rice grains (100 g) were suspended in a mixture of 6 mM KCl solution (0.045%, *w/v*) and ice (400 g in total; KCl solution: ice = 1:1), to ensure that the temperature of the mixture would remain below 50 °C during PEF treatment. A rectangular chamber {20 cm (L) × 10 cm (W) × 20 cm (H)} was used to hold rice suspensions during the PEF treatment.

The field strength of PEF treatment applied was 3 kV/cm. The pulse duration and frequency were 40 μs and 1 kHz, respectively. The PEF treatment was conducted on rice grains from 50 to 300 pulses. The initial and final temperatures and conductivity of samples were measured using a handheld thermocouple (Atkins AquaTuff 351, Cooper-Atkins Corp., Middlefield, CT, USA) and a TYPE 700 conductivity meter (CHEMTRIX, Inc., Hillsboro, OR, USA) respectively. Rice grains were drained after PEF treatment and immediately frozen using liquid nitrogen. Then the frozen samples were freeze-dried using a freeze-drier (In-Home HarvestRight, Harvest Right Inc., North Salt Lake, UT, USA) for further analysis. A portion of dry rice grains were ground using a homogenizer at 15,000 rpm for 3 min, and the milled rice flour was passed through a 125 μm sieve and collected. All samples were prepared in triplicate for each PEF treatment.

### 2.3. Porosity of Glutinous Rice Grains

The porosity of rice grains (*p*, %) was determined based on bulk density (*ρ_b_*, g/mL) and true density (*ρ_t_*, g/mL) according to Bhattacharya et al., [19] with some modifications. The bulk density was determined by using a 25 mL pre-calibrated graduated cylinder. An aliquot (10 g) of glutinous rice grains were loaded into the graduated cylinder, and the cylinder was gently tapped using a spatula for 20 s to adjust the clear top level of grains; *ρ_b_* was calculated by the ratio of mass of rice grain and volume occupied.

The true density was measured by the kerosene displacement method. An aliquot (10 g) of glutinous rice grains were gradually added in the graduated cylinder which was filled with a predetermined volume of kerosene (10 mL). The cylinder was mildly shaken on a L-RT30C platform rotator (United Products and Instrument Inc., Dayton, NJ, USA) while adding the rice, to vent the air bubbles adequately. *ρ_t_* was calculated by the ratio of mass of rice grain and volume of kerosene displaced. All samples were tested at least three times each.

The porosity (*p, %*) was calculated from *ρ_b_* and *ρ_t_* according to the Equation (1):(1)$$p(\%)\;=\;\frac{\rho_{t}-\rho_{b}}{\rho_{t}}\times 100$$

### 2.4. Scanning Electron Microscopy (SEM) Observation of Rice Grain Surface

Freeze dried rice grain samples were equilibrated overnight in a desiccator. Samples were fixed on aluminum studs using a double-sided adhesive tape, and then were sputter-coated with gold using a TED PELLA Cressington Sputter Coater (TED PELLA, Inc., Redding, CA, USA). The surface microstructure of glutinous rice grain samples with or without PEF treatment was observed using a JCM-6000 microscope (JEOL USA, Inc., Pleasanton, CA, USA). The sputter coated samples were transferred to microscope mount at an acceleration voltage of 15 kV, and observed at a magnification of 500×.

### 2.5. Pasting and Paste Properties of Glutinous Rice Flour

The pasting characteristics were measured and calculated according to Hagenimana and Ding [20] with some modifications. Pasting characteristics of rice flour and PEF treated rice flour were determined using an AR1500N rheometer (TA Instruments, New Castle, DE, USA). A rice flour suspension (7.5%, *w/w*) was prepared by mixing 0.6 g ground dry rice flour with distilled water. A portion of sample suspension was loaded onto sample plate and kept for 2 min at 25 °C to equilibrate. Time and temperature modes were utilized using a cone plate geometry system (40 mm diameter, 0.5° cone angle, and 1.000 mm gap). Test was performed at a constant shear rate of 50 rpm/min after equilibration. The suspensions were heated from 25 °C to 95 °C at a heating rate of 5 °C/min. They were maintained at 95 °C for 120 s, cooled to 25 °C at the rate 5 °C/min, and held at 25 °C for 60 s to develop the final paste viscosity. Data were collected every 10 s during the measurement. All measurements were done in triplicate. Peak viscosity was determined based on the maximum viscosity during pasting, and the final viscosity was obtained during the cooling stage. Breakdown viscosity was calculated according to the difference between the peak viscosity and minimum viscosity during pasting. Setback viscosity was calculated based on the difference between the maximum viscosity during cooling and the peak viscosity during pasting. Pasting temperature was determined based on the onset point of viscosity increase during the heating.

### 2.6. Differential Scanning Calorimetry (DSC)

The thermal analysis of the gelatinization behavior of rice flour with or without PEF treatment was measured using the Q2000 Modulated Differential Scanning Calorimeter (DSC, TA Instruments, New Castle, DE, USA) according to Qiu, et al. [21]. Well-mixed 2.5 ± 0.1 mg of rice flour powder samples and 7.5 µL of water were hermetically sealed in an aluminum pan. The prepared pans were heated from 20 to 130 °C at a heating rate of 10 °C/min under a continuous flow (60 mL/min) of dry N_2_ gas. The cooked pans were then cooled to 20 °C at the same rate. The onset, peak and conclusion temperatures (*T*_o_, *T*_p_ and *T*_c_) of gelatinization and gelatinization enthalpy (Δ*H*_gel_) were analyzed using TA Universal Analysis Q2000 software. Each test was performed for three times and the average values with standard deviation (SD) were calculated.

### 2.7. Fourier Transform Infrared Spectroscopy (FTIR)

Thirty (30) mg of rice flour was analyzed using the IRAffinity-1S Fourier transform infrared spectroscopy (FTIR) unit (SHIMADZU Corp. Japan) equipped with a Quest attenuated total reflectance (ATR) accessory (Specac Company, Orpington, UK) at the wavelength from 400 to 4000 cm^−1^. Measurements were performed at room temperature and 47% relative humidity. Signal averages were obtained for 32 scans at a resolution of 4 cm^−1^. An interactive baseline correction procedure was applied.

### 2.8. X-ray Diffraction (XRD)

The crystalline study of rice flour samples with or without PEF treatment was carried out using a Rigaku SmartLab X-Ray Diffractometer (Rigaku Denki Co., Ltd., Tokyo, Japan) operating at 40 kV and 25 mA, producing Cu/Kα radiation of 0.154 mm wavelength [22]. The freeze-dried samples were stored in desiccators overnight before testing. Diffractograms were obtained by scanning from 4° (2θ) to 40° (2θ) at an increment of 0.5°/min, a step size of 0.02°, a divergence slit width of 1°, a receiving slit width of 0.02 mm, and a scatter slit width of 1°. Each sample was measured twice.

### 2.9. ^13^C Cross-Polarization Magic Angle (CP/MAS) Nuclear Magnetic Resonance (NMR) Spectroscopy

The molecular orders of starch molecules in rice flour were investigated using solid-state NMR spectra [23]. The solid state ^13^C NMR was performed on a Bruker DRX 300 MHz system (Bruker Corp., Billerica, MA, USA), incorporated with a 7 mm cross-polarization magic angle (CP/MAS) probe. The rice flour samples were packed into a 6 mm diameter (o.d.) zirconium oxide rotor and spun at the right angle. The zirconium rotor was spun at the magic angle at 7000 cycles per second. The contact pulse was 1 ms, and the 90° high power pulse width was 5.05 µs. The relaxation delay between scans was set at 2 s, and 6000 scans were performed. Acquisition time was 16.4 ms, and two-pulse phase modulation (TPPM) decoupling was applied for data analysis.

### 2.10. Data Analysis

The reported results were obtained using the SAS system (Version 9.1 for Windows, SAS Institute Inc., Cary, NC, USA). Analysis of variance (ANOVA) was used to determine significant differences between the results. The Duncan’s test was used to separate the means with a significance level of 0.05.

## 3. Results and Discussion

### 3.1. Proximate Analysis, Microstructure and Porosity of Glutinous Rice Grains

The proximate analysis data of native glutinous rice grains are shown in Table 1. Glutinous rice grain contained 83.1% total starch, and the starch mainly consisted of amylopectin (98.2%). Rice grain also contained a large amount of protein (8.4%) and negligible amount of fat and ash.

Temperature and electric conductivity of rice suspensions before and after PEF treatment are also reported in Table 1. The ice/KCl solution mixtures were added before PEF treatment as the lab-scale PEF unit was not equipped with cooling accessories. Temperature was determined to be 51.3 °C after 3 kV/cm, 300 pulses treatment. According to previous studies, the temperature of treated starch suspension was maintained below 50 °C during PEF treatment to control starch gelatinization [16,24]. The PEF unit and PEF treating parameters applied allowed the study of the starch properties while avoiding the effects of temperature-induced gelatinization. It was also found (see Table 1) that more pulses of PEF treatment generated higher suspension temperature, which also increased the electric conductivity due to the promotion of ions movement and ice melting.

Scanning electron microscopy (SEM) images of rice grain surface with or without PEF treatment are shown in Figure 1. For the microstructure of native glutinous rice grain (Figure 1a), compact structures and arrangements of rice components were mainly observed on the grain surface. The tightly packed structure with fewer interspaces on rice surfaces were also observed on the images of polished rice published in other studies [25]. Some incompact structures and flakes were found on rice grains as a result of adequate water absorption after 2 h soaking. It was reported that the rice starch and waxy rice starch particles were irregular polyhedral in shape with an average diameter range of 1–7 μm [17]. The border of flakes and pores of soaked rice grain was irregular and jagged in Figure 1a because of the arrangement of starch particles. In addition, shallow pores and cavities were found on the surface of rice grain which would act as channels that improved the interactions of water and internal rice components during the gelatinization treatment.

Well-defined and deeper pores were observed at the rice surface (Figure 1b–d) as the input energy of PEF treatment was raised. It was also found that the porosity of 3 kV/cm, 50 to 200 pulses PEF treated rice grains was increased from 7.29% to 8.65% compared to control (Table 2). Both true density and bulk density insignificantly (*p* < 0.05) increased after PEF treatment indicating that PEF treatment changed surface microstructure without altering the size, shape and integrity of rice grains. The pores on PEF treated rice grain surface were similar to the results of PEF treated potato slices reported by Ammar, et al. [26], which also stated that the porosity of PEF treated potato was 84% whereas the porosity of the control sample was 71%. However, the shape of pore or cavity on PEF treated rice grains was completely different from the pores on potato, which showed deformation and damage of the polyhedrally shaped cells after PEF treatment [26]. The constitution and structure of rice grain were much different than in potato chunks. Potato starch granules were embedded within the potato cell membrane which consisted of cellulose and pectin [27], and the PEF induced electroporation effects resulted in the pore-forming or breakdown in the potato cell membrane [20]. The surface of glutinous rice grain, in our current study, was composed of an aleurone layer and starchy endosperm [27]. Visible starch granules formed, tightly organized in the aleurone layer and endosperm during the growth of rice, similarly to other cereal starches [28]. The pores on the glutinous rice surface induced by PEF treatment could be utilized as channels during food processing which could eventually improve the quality of rice products. For instance, the interaction of glutinous rice grains and other ingredients, such as water, sugar, oil and food hydrocolloid, could be enhanced during the preparation of glutinous rice cake due to the increased porosity. The glutinous rice cake prepared with PEF pretreatment could have a smoother texture and more homogeneous distribution of flavor during food processing. Small protein fractions (8.4%, Table 2) were scattered throughout the surface at the peripheral outer layer of rice grain, mainly embedded in the spaces between starch granules. Therefore, the pores on rice grains could be induced by the structural changes of electroporation-sensitive proteins and starch molecules during PEF treatment.

The rice segments were observed to be incompact and loose after 300 pulses of PEF treatment (Figure 1d). Some grooves or aperture structures were found on the rice surface. Moreover, some starch particles were apparently larger and more well-defined than starch particles shown in the control sample. Granular surface pits in wheat and potato starches at the field strength of 30 and 40 kV/cm during PEF treatment were also reported in previous studies [15,16]. The porosity of 300 pulses PEF treated rice grains increased to 9.81%, significantly higher than the control (Table 2). Thus, PEF treatment was able to both change the microstructure of starch granules and the surface structure of glutinous rice grains.

### 3.2. Pasting and Paste Properties of Glutinous Rice Flour

The viscosity parameters of ground glutinous rice flour with or without PEF treatment are shown in Table 3. The peak viscosity of control and PEF-treated samples showed a corresponding decrease from 1484 to 1407, 1420, 1350 and 1172 Pa∙s, respectively when the number of PEF pulses increased from 50 to 300. This result was consistent with previous studies that higher field strength of PEF (30–50 kV/cm) treatment could decrease the peak viscosity of potato and maize starches compared to the control [24]. Peak viscosity reflects the swelling ability of the starch granules before their physical breakdown during the gelatinization process [24]. Tester and Morrison (1990) reported that amylopectin contributes to swelling degree, whereas the amylose and lipids inhibited swelling for native starch. During the gelatinization of glutinous rice flour, the swelling degree of starch granules should be enhanced because of PEF-induced electroporation or better hydration effects. Thus, the PEF treatment could facilitate the swelling and rapture of starch granules during the gelatinization, and the early point of granule rapture of starch granules would result in a lower peak viscosity during viscosity measurement of heating.

After starch suspension reached the maximum viscosity, starch granules began to rupture and break down which resulted in a viscosity decrease. The higher the intensity of PEF treatment, the larger values of breakdown and the slightly larger values of final viscosity were observed, as shown in Table 3. A significant increase (*p* < 0.05) in breakdown viscosity of PEF-treated rice flour could be due to larger swelling degree of the starch granules as well [21]. The value of breakdown viscosity was also utilized to analyze the corresponding viscosity change of starches to heating and shear thinning during gelatinization [29]. Higher values of breakdown viscosity suggested that the PEF-treated starch granules could be more resistant to mechanical and thermal treatment during cooking [30]. Therefore, PEF treated glutinous rice flour was able to swell to a lager extent which was more resistant to breakdown during cooking.

The setback value of rice flour significantly increased and pasting temperature of rice flour was almost unchanged when the PEF treating pulses were increased from 50 to 300 pulses. This result indicated that the PEF treatment improved the retrogradation of starch at the cooling stage of viscosity measurement [21], as a result of a larger extent of swelling during the pasting/heating process. The unchanged pasting temperature in this study contradicted previous results of PEF treated starch [16,17,24], which widely reported that PEF treatment decreased the pasting temperature of starch suspension. The significant amount of protein (8.4%) embedded in the starch matrix of the glutinous rice grains could maintain the beginning of swelling and gelatinization at the similar temperature point, and it contributed to the unchanged pasting temperature [30].

In summary, the peak viscosity of PEF treated rice flour was reduced with increasing treating pulses. The breakdown, final and setback viscosity of glutinous rice flour increased as the treated PEF pulses increased. These results indicated that PEF treated glutinous rice flour was more vulnerable to gelatinization, and more resistant to thermal treatment during viscosity measurement of heating due to the structural changes of starch/rice grains or loss of starch crystallization after PEF treatment.

### 3.3. Thermal Properties

Values of *T_o_*, *T_p_*, *T_c_* and Δ*H_gel_*, calculated from the DSC measurements are shown in Table 4. Statistical analysis in Table 4 indicated that *T_o_* and Δ*H_gel_* values of PEF treated rice flour decreased as the PEF treating pulses increased from 50 to 300 pulses. There were no significant differences among the *T_p_* and *T_c_* values of various rice flour samples and the control, which only showed a slight decrease in the 300 pulses, PEF treated glutinous rice sample. The gelatinization temperatures and enthalpy could be related to the degree of crystallinity and amylopectin structure of starch granules [27]. This result agreed well with previous studies [16] that 806 μs of PEF treatment at high field strength of 40 kV/cm could decrease the gelatinization temperature from 65.15 °C to 64.61 °C. Hong, et al. [31] also reported that adequate input energy of PEF treatment could change the internal structure of starch granules resulting in less attractive forces in the amorphous region and weaker intra-molecular hydrogen bonds, making the gelatinization happen at lower temperature.

The gelatinization enthalpy indicates the energy that amylopectin requires for the phase transformation during DSC measurement. Many previous studies reported that high field strength (30–50 kV/cm) and short pulses of PEF treatment changed the gelatinization enthalpy of native starches. According to Han, et al. [16], DSC calculated gelatinization enthalpy of tapioca starch remarkably decreased from 16.06 J/g to 8.87 J/g, 4.82 J/g and 3.80 J/g when the field strength increased from 0 to 30, 40 and 50 kV/cm respectively. In addition, Δ*H_gel_* of PEF treated waxy rice starch was reported to decrease from 11.8 J/g to 9.8 J/g with increasing field strength from 0 to 50 kV/cm [17].

The lower field strength of PEF of 3 kV/cm and 300 pulses (pulse width = 40 μs), in this study, was confirmed to induce similar changes in thermal properties of glutinous rice flour.

### 3.4. X-ray Diffraction (XRD) Pattern

X-ray diffraction (XRD) was utilized to investigate the amorphous/crystalline state of the control and PEF treated rice sample powders. Native glutinous rice flour showed a typical A-type XRD pattern with clear peaks at 2θ close to 15.8°, 17.1°, 18.3° and 23.1° (Figure 2), according to the crystalline pattern of cereal starches reported in previous studies [22,27]. This result agreed with the A-type pattern of normal rice flour and waxy rice flour displayed in X-ray diffraction analysis [27].

The XRD patterns of 50 pulses and 100 pulses PEF-treated rice grain were similar to the control sample. The peak intensity at 17.1° of 100 pulses PEF-treated rice was slightly weaker than control. However, the XRD crystallization peaks of rice samples with 200 to 300 pulses were much smaller and smoother than the control. The peak intensity at 23.1°of 300 pulses PEF-treated rice sample was weaker and the peak pattern was much smoother than other rice samples. Larger PEF treatment intensity led to a diminishing XRD pattern which indicated a decrease in the XRD peak intensity and relative crystallinity of rice grain. It was reported that gelatinized starch granules showed a diffuse pattern, which was an indication of changing from crystalline to amorphous structure, suggesting the substantial loss of A-type crystallinity in these starch granules because of gelatinization [27]. According to previous studies, PEF treatment could destroy the crystalline region of starches, by imposing the electric field of higher energy on starch molecular chains than the energy required for the formation of starch hydrogen bond [15,17]. Therefore, the XRD results in this study indicated that partial gelatinization had occurred in glutinous rice samples after 200 and 300 pulses PEF treatment. The XRD results correlated well with the changes in endothermic enthalpy of gelatinization given in Table 4. Similar XRD results showed that crystalline patterns of the corn, potato, and tapioca starches were partially disrupted after 30–50 KV/cm PEF treatment [15,16,24].

This study confirmed that low field strength (3 kV/cm) and adequate number of treating pulses (200–300 pulses) were able to induce partial gelatinization of starch in whole glutinous rice grains, and that 200 to 300 pulses of 3 kV/cm PEF treatment were able to induce structural changes of rice grains as well as microstructure changes of starch molecules.

### 3.5. Fourier Transform Infrared (FTIR) Spectroscopy Results

FTIR spectra were determined to investigate the structural and conformational changes of glutinous rice flour after PEF treatment (Figure 3). The four PEF treated samples and native rice flour had similar peaks position at the regions of 4000 to 2500 cm^−1^ and 800 to 400 cm^−1^. The peaks of rice flour at 3250 and 2929 cm^−1^ were the assignments of O–H and –CH_2_ stretching, respectively [32]. Rice with 200 and 300 pulses PEF treatment showed apparently stronger intensities at the peaks of 3250 and 2929 cm^−1^ compared to control (Figure 3). It was reported that the hydrated rice flour samples showed more pronounced peaks at 3700–3000 cm^−1^ and 2250–2000 cm^−1^ (Figure 3B), which was an indication of absorption by excess water or hydration [33]. Thus, this result confirmed that 50, 100, 200 and 300 pulses of PEF enhanced the hydration of rice grains compared to control rice grains soaked in 0.045% KCl solution. Bands at 784, 738, 578 and 524 cm^−1^ were due to the skeletal mode vibrations of the pyranose ring in the glucose unit, which was confirmed with both Raman and infrared spectra of starch [32]. The FTIR spectra of different rice samples at those regions were similar in Figure 3A. This result indicated that PEF treatment did not cause any new chemical group or molecular dissociation on starch molecules of glutinous rice flour.

Bands at the region of 1700 to 1200 cm^−1^ were reported to be associated with minor functional components (protein, lipid) in the rice flour (Warren, et al., 2016). At the region of 1700 to 1200 cm^−1^ in the overlay spectra (Figure 3C), a single peak and broad twin-peak of rice flour with or without PEF treatment were found at the wavenumbers of 1687, 1653and 1649 cm^−1^. The shapes of those peaks became sharper and more distinct after PEF treatment. According to Kong and Yu, [34], bands at 1690, 1680–1675, 1642–1624 cm^−1^ were assigned to β-sheets structure, and bands at 1680 and 1666 cm^−1^ were assigned to β-turns structure, 1655–1645 cm^−1^ to random coils structure, and 1655 and 1650 cm^−1^ to α-helices structure. Therefore, the shape changes of bands at 1687, 1653 and 1649 cm^−1^ were related to the structure change of inter-molecular β-sheets and α-helices of rice protein, indicating protein unfolding and secondary structure change after PEF treatment [35]. Moreover, new peaks of PEF treated samples were identified at the wavenumbers of 1558, 1531, 1522 and 1507 cm^−1^, respectively. The band near the wavenumber of 1550 cm^−1^ corresponds to C=O stretching in free carboxyl groups (COO–) [36]. The band at 1530 is related to amide II bond and N–H bending band [36]. These results suggested that PEF treatment influenced the vibration of C=O and the deformation of N–H, which was an also indication of rice protein unfolding after PEF processing. Therefore, the PEF treatment applied in this study was able to change the β-sheets and α-helices structures of the secondary structure of rice protein, and induced protein unfolding in whole rice grains.

The FTIR bands of starch depicted at the wavenumbers of 1300–800 cm^−1^ indicated the stretching vibration of functional groups of C-O, C-C, C-O-H and of the glycosidic bond of C-O-C in starch [33]. In Figure 3D, the FTIR peaks of rice flour samples with or without PEF treatment showed clear and similar peaks at the wavenumbers of 1153, 1078, 1024, 995, 929, 860 cm^−1^. However, the intensities of each peak at this region dramatically increased as the pulses of PEF treatment were raised from 200 to 300 pulses, indicating the partial gelatinization of starch granules during PEF treatment [27]. This finding agreed well with previous studies of PEF treated starch, describing that those PEF-induced changes were similar to the hydration reaction caused by starch gelatinization [37].

### 3.6. Nuclear Magnetic Resonance (NMR)

The solid state ^13^C CP/MAS NMR spectra of rice flour samples with or without PEF treatment are presented in Figure 4. The NMR spectra were measured to investigate the molecular orders of starch molecules, and our data showed that the spectra of rice flour were similar to that of corn starch samples stated in previous studies [38]. It has been reported that signals at 99 ppm to 104 ppm were correlated to C-1 resonance peak which was related with V-type single helix of amylose or double helices of short chain amylopectin; signals at 81 ppm to 84 ppm corresponded to C-4 resonance peak which indicated starch amorphous fractions; and signals at 59 ppm to 62 ppm were assigned to hydroxymethyl carbon-6 structure in hexopyranoses [38]. The most pronounced signal at 70–73 ppm was associated with C-2, 3, 5 in hexopyranoses [38]. In this study, there were no changes on peak shape and no emerging peak in NMR spectra, which agreed well with the FTIR results. However, it was clear that the peaks at 101.5, 73.1 and 62.8 ppm were significantly shifted to higher resonance after PEF treatments (Figure 4). According to Cheetham and Tao [38], the shift of resonance peak and shoulder is due to the low field resonance of non-crystalline material. In our current study, those peak shifts were attributed to the enhancement of starch hydration after PEF treatment, showing that higher PEF intensity promotes the peaks’ shift to higher resonances. In addition, PEF treatment induced starch rearrangement of glutinous rice grains, and thus fortified starch amorphous structures [38].

## 4. Conclusions

This study applied 50, 100, 200 and 300 pulses of PEF treatment at moderate field strength (3 kV/cm) to whole glutinous rice grains. The physicochemical properties, microstructure and gelatinization behaviors of glutinous rice grains prepared with or without PEF treatments were studied. Results have shown that well-defined and deeper pores were embedded in the rice surface, and porosity was increased from 7.3% to 9.8% as the pulses of PEF treatment increased from 0 to 300 pulses. PEF-treated rice flours exhibited a significant peak viscosity value decrease during gelatinization as a result of larger swelling degree. Thermal properties, X-ray diffraction results and infrared, NMR spectra of glutinous rice flours with or without PEF treatment, indicated that PEF was able to induce partial gelatinization and better hydration of whole glutinous rice grains, which made the PEF treated starch granules more accessible to water during gelatinization. FTIR results showed that PEF treatment changed inter-molecular β-sheets and α-helices structures of rice protein, indicating changes in protein unfolding of secondary structures.

Moderate PEF treatment (3 kV/cm, 50–300 pulses) changed the microstructure of starch and protein components in whole, water-soaked glutinous rice grains and induced structural surface modifications, which were only reported in starch particles/starch powder at high field strength (30–50 kV/cm) of PEF treatment in previous studies. Such modifications will allow the manufacture of rice products with better quality and shorter processing times where faster hydration, lower gelling temperature and uniform absorption of ingredients are required.

Commercial glutinous rice products, such as rice cake, are composed of glutinous rice with other ingredients including sugar, oil and stabilizers, therefore studying their interactions during and after PEF treatment would be required to assess industrial applications, which were not investigated in this study. Thus, the gelling behaviors, gel textural properties and retrogradation behaviors of glutinous rice products prepared with PEF treatment warrant further research.

## Figures and Tables

**Figure 1 foods-10-00395-f001:**
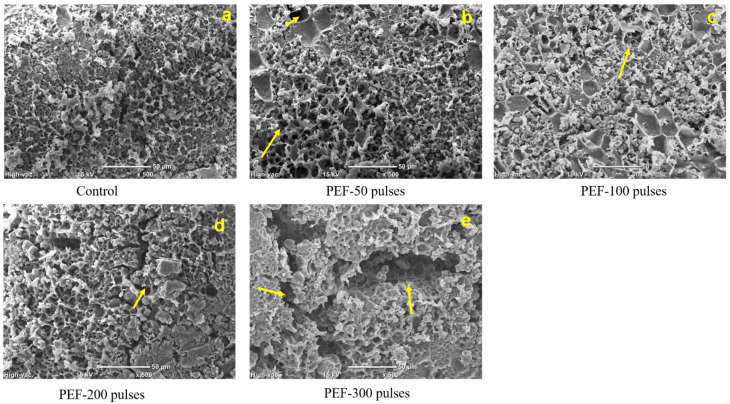
Scanning electron microscopy (SEM) images of pre-soaked glutinous rice grain surface without PEF treatment (Control, **a**), and with 50 (**b**), 100 (**c**), 200 (**d**), and 300 (**e**) pulses of PEF treatment. Arrows indicate the pores or cavity on PEF treated rice grain surface. The SEM images captured for all samples provided a ×500 magnification and used scale bar of 50 μm.

**Figure 2 foods-10-00395-f002:**
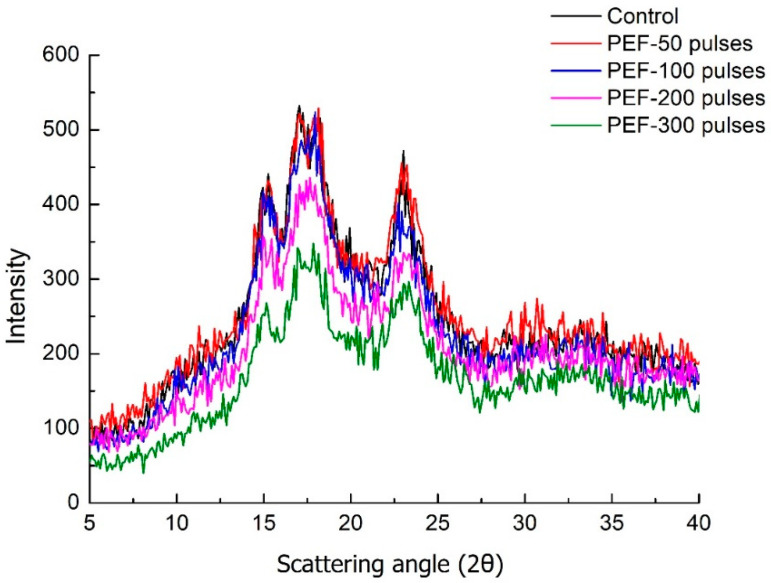
Overlay X-ray diffractograms of control and PEF-treated glutinous rice flour at the field strength of 3 kV/cm for 50, 100, 200 and 300 pulses.

**Figure 3 foods-10-00395-f003:**
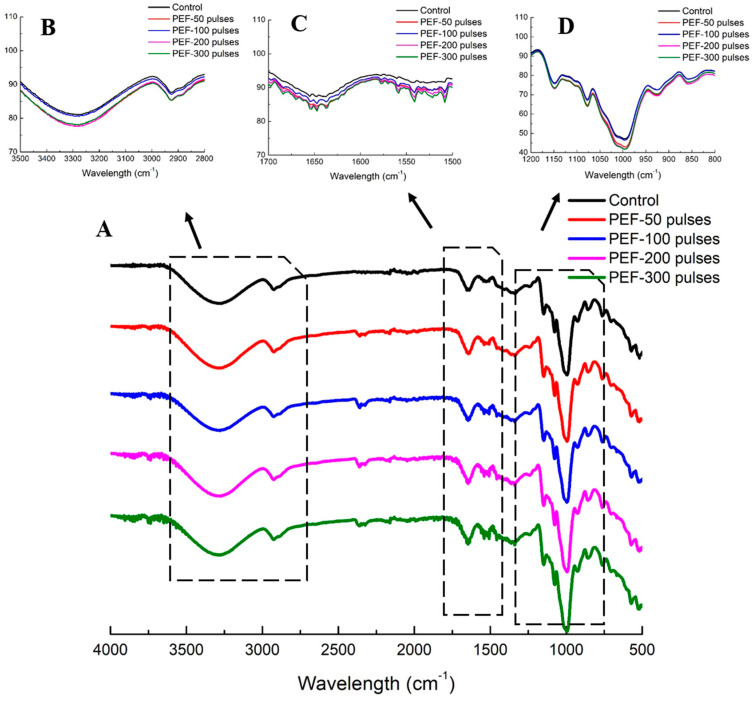
Separated and overlay Fourier transform infrared (FTIR) spectra (**A**) of control and PEF treated glutinous rice flour at the field strength of 3 kV/cm for 50, 100, 200 and 300 pulses. The overlay spectra have been zoomed to show the regions of 3500–2800 cm^−1^ (**B**), 1700–1500 cm^−1^ (**C**) and 1200–800 cm^−1^ (**D**).

**Figure 4 foods-10-00395-f004:**
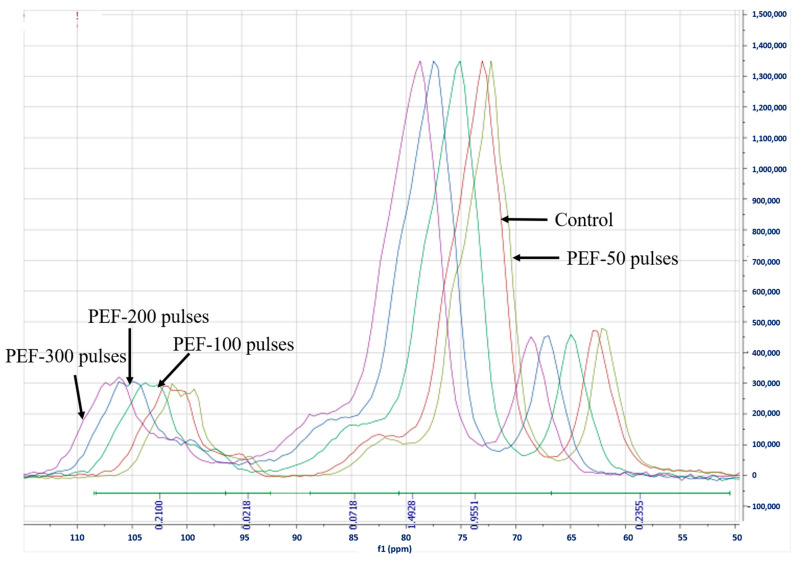
^13^C cross-polarization magic angle (CP/MAS) nuclear magnetic resonance (NMR) spectra of control and PEF treated glutinous rice flour at the field strength of 3 kV/cm for 50, 100, 200 and 300 pulses.

**Table 1 foods-10-00395-t001:** Proximate composition of glutinous rice grains (wt.%, dry weight basis), and the temperature of rice grains suspensions before and after pulsed electric field (PEF) treatment *.

	Moisture	Ash	Crude Protein	Crude Fat	Total Starch	Amylose
**Glutinous rice grains**	12.45 ± 0.22	0.24 ± 0.02	8.44 ± 0.17	0.50 ± 0.02	83.07 ± 2.39	1.85 ± 0.27
	Treatment	Control	PEF-50 pulses	PEF-100 pulses	PEF-200 pulses	PEF-300 pulses
	Initial *T* (°C) ^#^	0.7 ± 0.0	0.7 ± 0.1	0.9 ± 0.0	1.1 ± 0.2	1.2 ± 0.1
	Final *T* (°C)	1.3 ± 0.0	3 ± 0.7	17.7 ± 0.3	30.8 ± 0.9	51.3 ± 1.4
	Initial *θ* (μS/cm)	250 ± 1.0	250 ± 1.0	260 ± 1.7	250 ± 0.7	267 ± 3.0
	Final *θ* (μS/cm)	250 ± 0.3	250 ± 3.1	300 ± 3.0	390 ± 1.0	400 ± 2.5

* Data are means of duplicate ± SD. ^#^
*T*, temperature (°C); *θ*, electric conductivity (μS/cm).

**Table 2 foods-10-00395-t002:** Density and porosity of soaked glutinous rice grains with or without PEF treatment.

Treatment	Bulk Density (*ρ_b_*, g/mL)	True Density (*ρ_t_*, g/mL)	Porosity (*p* (%) = (*ρ_t_* − *ρ_b_*)/*ρ_t_* × 100)
Control	1.50 ± 0.02 ^a^	1.39 ± 0.01 ^a^	7.29 ± 0.01 ^a^
PEF-50 pulses	1.50 ± 0.04 ^a^	1.38 ± 0.01 ^a^	7.45 ± 0.02 ^a^
PEF-100 pulses	1.50 ± 0.00 ^a^	1.38 ± 0.00 ^a^	7.44 ± 0.00 ^a^
PEF-200 pulses	1.52 ± 0.13 ^a^	1.38 ± 0.03 ^a^	8.65 ± 0.04 ^a,b^
PEF-300 pulses	1.55 ± 0.01 ^a^	1.40 ± 0.00 ^a^	9.81 ± 0.01 ^b^

Mean values ± standard deviation of triplicates; values in the same column with different superscripts are significantly different at *p <* 0.05 level.

**Table 3 foods-10-00395-t003:** Viscosity parameters of glutinous rice flour with or without PEF treatment measured in the rheological test of time and temperature modes.

Treatment	Peak Viscosity (Pa·s)	Breakdown (Pa·s)	Final Viscosity (Pa·s)	Setback (Pa·s)	Pasting Temperature (°C)
Control	1484 ± 1 ^a^	153 ± 1 ^a^	1812 ± 4 ^a^	328 ± 7 ^a^	68.1 ± 0.0 ^a^
PEF-50 pulses	1407 ± 4 ^a^	155 ± 3 ^a^	1802 ± 3 ^a^	395 ± 0 ^b^	68.2 ± 0.0 ^a^
PEF-100 pulses	1420 ± 01 ^a^	159 ± 1 ^a^	1811 ± 4 ^a^	391 ± 1 ^b^	68.1 ± 0.1 ^a^
PEF-200 pulses	1350 ± 10 ^a,b^	198 ± 2 ^a,b^	1805 ± 3 ^a^	455 ± 6 ^c^	68.2 ± 0.0 ^a^
PEF-300 pulses	1172 ± 3 ^b^	290 ± 3 ^b^	1827 ± 5 ^a^	655 ± 7 ^d^	68.1 ± 0.1 ^a^

Mean values ± standard deviation of triplicates; values in the same column with different superscripts are significantly different at *p <* 0.05 level.

**Table 4 foods-10-00395-t004:** Differential scanning calorimetry (DSC) measurements for gelatinization properties of glutinous rice flour with or without PEF treatment ^#^.

Treatment	*T_o_* (°C) *	*T_p_* (°C)	*T_c_* (°C)	Δ*H_gel_* (mJ/g)
Control	61.31 ± 0.02 ^a^	66.23 ± 0.26 ^a^	73.89 ± 0.05 ^a^	−9.29 ± 0.13 ^a^
PEF-50 pulses	61.27 ± 0.33 ^a^	66.25 ± 0.05 ^a^	72.88 ± 0.14 ^a^	−8.93 ± 0.08 ^a^
PEF-100 pulses	59.70 ± 0.27 ^a,b^	66.09 ± 0.21 ^a^	73.12 ± 0.09 ^a^	−8.35 ± 0.24 ^a,b^
PEF-200 pulses	57.75 ± 0.41 ^b^	66.72 ± 0.13 ^a^	73.45 ± 0.03 ^a^	−7.93 ± 0.11 ^b^
PEF-300 pulses	57.53 ± 0.17 ^b^	65.32 ± 0.70 ^a^	73.92 ± 0.22 ^a^	−7.28 ± 0.39 ^b^

^#^ Mean values ± standard deviation of triplicates; values in the same column with same superscripts do not differ significantly at *p <* 0.05 level. * *T_o_*, onset temperature; *T_p_*, peak temperature; *T_c_*, conclusion temperature; Δ*H_gel_*, enthalpy of gelatinization.

## Data Availability

The data presented in this study are available upon the request from the corresponding author. The data are not publicly available due to the confidential agreement with the sponsor.

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
