# Peer review of "Changes in the Glutinous Rice Grain and Physicochemical Properties of Its Starch upon Moderate Treatment with Pulsed Electric Field"

_foods, 2021, doi:10.3390/foods10020395_

Round 1

Reviewer 1 Report

Review - Manuscript ID: Foods-1101910

Changes in the glutinous rice grain and physicochemical properties of its starch upon moderate treatment with Pulsed Electric Field

General remarks

The work presented addresses an interesting subject. The objectives of the study are perfectly introduced by a clear and precise bibliographic part. The presentation of the experimental device and the methods used is very complete and relevant. In general, the presentation and discussion of the main results obtained is correct. However, as noted in detail below, a number of improvements must be considered. Finally, reading the conclusions formulated by the authors, the reader is frustrated by the few perspectives given to this work?

Minor remarks

Figures 1, 2 and 3: In general, the quality of the figures presented must be improved. Indeed, the reader has great difficulty in acquainting himself with the information presented with precision.

By way of example, for fig. 1, the observation scales are illegible and the reader has difficulty understanding what the arrows positioned in the various images correspond to.

Table 2 (page 5): The format of table 2 must be corrected on several levels:

  • First of all, if the authors wish to propose additional captions, they must still correctly reference the additional information. So, I suggest that lines 213-214 be deleted as these are redundant from the title of the table.
  • The information listed in line 215 should be referenced with a different symbolism; furthermore I suggest that the units used be placed in parentheses.
  • Finally, it would be wise to delete the fourth row of the table which is perfectly redundant with the preceding one.

Lines 238, 245, 249, 272: The authors refer respectively in these different lines to Figures 1a, 1b-d and 1d. In all consistency, it would be desirable for Figure 1 (page 6) to use this same referencing system (a, b, c, d and e) in order to avoid any confusion.

Table 3 (page 8): In order to respect a certain homogeneity of presentation, the title of the last column should correspond to “Pasting Temperature (° C)”.

Table 4 (page 9): Authors should choose different markers for additional footnotes (lines 350-353). At the same time, it does not seem necessary to recall the units used for qualified T0, Tp, Tc and ΔH-gel (lines 350 and 351) because this is already done in the first row of the table.

Line 336: The authors refer to a table 2B; isn't that Table 4?

Line 413: The authors refer to Fig. 2; isn't that Fig. 3?      

References (page 14): Following a quick review of the references presented, I invite the authors to correct now the format of these insofar as it does not completely respect the standard of the journal.

Reviewer 2 Report

The work is very interesting. The procedures and results obtained are very well detailed.
To further improve the quality of this work you can find some suggestions/recommendations:

Introduction

-Lines 52 – 53. It is well known that the viscoelastic properties depend mainly on the amylose and amylopectin content. Some lines could be included detailing the behavior of these polysaccharides, when subjected to thermal and electrical changes in this case.

Results

-Line 205. 3.1 Proximate analysis, microstructure and porosity of glutinous rice grains. The effect of the Temperature and electric conductivity of rice suspensions before and after PEF treatment should be explained in more detail. What happens to the structure of the sample and how it starts to be affected by these variables.

-Figure 1. SEM images. Magnification and scale should be included within the images for better visualization.

-Line 249. Respect to the density and porosity of soaked glutinous rice grains. In the discussion of this section could mention what are the benefits of having lower or higher porosity, as well as density and viscosity for food applications.

Conclusions

It would be interesting to include a few lines highlighting the importance of these results for practical purposes (i.e. food industry).

Reviewer 3 Report

In this study, the authors investigate the effects of Pulsed Electric Field on glutinous rice grains. In particular, they aim at using moderate field strength (3 kV/cm) with longer treating time and/or increasing the pulse number to obtain conditions more suitable to industrial application.

The study is well designed and conducted. PEF 50 up to 300 pulses are applied to the samples and the temperature monitored. Different techniques are used: SEM is used to study the porosity level as well as sample density, thermal properties were studied by DSC measurements, X diffraction to study gelatinization level as a function of the number of pulses, FTIR and solid state 13CNMR to study the structural and conformational changes of the rice samples. Encouraging conclusions are reached as concerns porosity, viscosity and thermal properties of the samples after the treatment.

The criticisms raised by this reviewer are as follows:

  1. In the Introduction and/or in the Conclusions section, the authors should address in more depth the real improvement of the quality that is in turn strictly connected to its application by food industry.
  2. the composition of glutinous rice grains is reported in Table 1. What about the same composition as concerns nutritional properties after the PEF-pulses treatments?
  3. SEM analysis. Figure 1 does not report the different letters, nor image resolution: that is the dimension the bars are referred to. In particular, it is not clear if Figure 1a is referred to the control sample after 2h soaking or before, otherwise line 242 not shown has to be added.
    1. Figure 2 caption: angle and not angel!
  4. NMR analysis. Figure 4 is not well distinguishable. Both the ppm values and the spectra are not quite visible.

Which the procedure adopted by the authors to assign the resonances? In particular at Line 461 they refer to a literature study, but this is not shown in the Reference section [38].  Was it at the same field, temperature, experimental condition?

Line 470: higher resonance is as a matter of facts low field resonance and/or higher frequency.

  1. A Section reporting the Limitation of the study should be recommended.

Author Response

Response to Reviewer 3 Comments:

In this study, the authors investigate the effects of Pulsed Electric Field on glutinous rice grains. In particular, they aim at using moderate field strength (3 kV/cm) with longer treating time and/or increasing the pulse number to obtain conditions more suitable to industrial application.

The study is well designed and conducted. PEF 50 up to 300 pulses are applied to the samples and the temperature monitored. Different techniques are used: SEM is used to study the porosity level as well as sample density, thermal properties were studied by DSC measurements, X diffraction to study gelatinization level as a function of the number of pulses, FTIR and solid state 13CNMR to study the structural and conformational changes of the rice samples. Encouraging conclusions are reached as concerns porosity, viscosity and thermal properties of the samples after the treatment.

The criticisms raised by this reviewer are as follows:

  1. In the Introduction and/or in the Conclusions section, the authors should address in more depth the real improvement of the quality that is in turn strictly connected to its application by food industry.

Response 1: Thanks for your suggestions. We have added the contents in the Introduction and Conclusion Sections as follows:

In Line 80, “Therefore, PFE treatment can be utilized as a novel processing method to potentially develop glutinous rice products with improved textural properties and stability.”

In Line 520, “Such modifications will allow the manufacture of rice products with better quality and shorter processing times where faster hydration, lower gelling temperature and uniform absorption of ingredients are required.”

  1. the composition of glutinous rice grains is reported in Table 1. What about the same composition as concerns nutritional properties after the PEF-pulses treatments?

Response 2: PEF treatment could induce slight variation on the contents of moisture, crude protein, and total starch as a minor fraction of glutinous rice grain may leach out during the treatment. We did not evaluate the nutritional properties and rice composition after PEF treatment as we are interested in the physical attributes and microstructure properties of the PEF treated rice grains. Thus, we did not report the nutritional properties after the PEF-pulses treatments in this study.

  1. SEM analysis. Figure 1 does not report the different letters, nor image resolution: that is the dimension the bars are referred to. In particular, it is not clear if Figure 1a is referred to the control sample after 2h soaking or before, otherwise line 242 not shown has to be added. a. Figure 2 caption: angle and not angel!

Response 3: We have revised all figures that are included in the manuscript based on your suggestion. The caption of Fig. 1 was also revised to “SEM images of pre-soaked glutinous rice grain surface without PEF treatment (Control, a), and with 50 (b), 100 (c), 200 (d), and 300 (e) pulses of PEF treatment. Arrows indicate the pores or cavity on PEF treated rice grain surface. The SEM images captured for all samples provided a ×500 magnification and used scale bar of 50 μm.”

Figure 2 has been corrected, thank you for the catching the error (angle not angel).

  •  
  1. NMR analysis. Figure 4 is not well distinguishable. Both the ppm values and the spectra are not quite visible.

Which the procedure adopted by the authors to assign the resonances? In particular at Line 461 they refer to a literature study, but this is not shown in the Reference section [38]. Was it at the same field, temperature, experimental condition?

Line 470: higher resonance is as a matter of facts low field resonance and/or higher frequency.

Response 4: Figure 4 has been improved by redoing the labeling and scales. The reference citation and associated explanation have been added (see below).

According to Cheetham, & Tao, (1998), the native corn starch samples, with the apparent amylose content varied from 0% to 84%, were measured using the solid state 13C CP/MAS NMR at the magic-angle spinning rate of 2.5 KHz and decoupling field of 61 KHz. Spectra are referenced to external Me4Si via the low field resonance of adamantane (38.6 ppm) at room temperature. The spectra of corn starch (following figure) agree with the rice flour spectra presented in our study (Fig. 4).

The following contents have been added in this section, “According to Cheetham, & Tao (38), the shift of resonance peak and shoulder is due to the low field resonance of non-crystalline material. In our current study,”

Fig. 4.

  1. A Section reporting the Limitation of the study should be recommended.

Response 5: the following sentences have been added to the Conclusion Section, “Commercial glutinous rice products, such as rice cake, are composed of glutinous rice with other ingredients including sugar, oil and stabilizers, therefore studying their interactions during and after PEF treatment would be required to assess industrial applications, which were not investigated in this study. Thus, the gelling behaviors, gel textural properties and retrogradation behaviors of glutinous rice products prepared with PEF treatment warrant further research.”

Response to Reviewer 3 Comments:

In this study, the authors investigate the effects of Pulsed Electric Field on glutinous rice grains. In particular, they aim at using moderate field strength (3 kV/cm) with longer treating time and/or increasing the pulse number to obtain conditions more suitable to industrial application.

The study is well designed and conducted. PEF 50 up to 300 pulses are applied to the samples and the temperature monitored. Different techniques are used: SEM is used to study the porosity level as well as sample density, thermal properties were studied by DSC measurements, X diffraction to study gelatinization level as a function of the number of pulses, FTIR and solid state 13CNMR to study the structural and conformational changes of the rice samples. Encouraging conclusions are reached as concerns porosity, viscosity and thermal properties of the samples after the treatment.

The criticisms raised by this reviewer are as follows:

  1. In the Introduction and/or in the Conclusions section, the authors should address in more depth the real improvement of the quality that is in turn strictly connected to its application by food industry.

Response 1: Thanks for your suggestions. We have added the contents in the Introduction and Conclusion Sections as follows:

In Line 80, “Therefore, PFE treatment can be utilized as a novel processing method to potentially develop glutinous rice products with improved textural properties and stability.”

In Line 520, “Such modifications will allow the manufacture of rice products with better quality and shorter processing times where faster hydration, lower gelling temperature and uniform absorption of ingredients are required.”

  1. the composition of glutinous rice grains is reported in Table 1. What about the same composition as concerns nutritional properties after the PEF-pulses treatments?

Response 2: PEF treatment could induce slight variation on the contents of moisture, crude protein, and total starch as a minor fraction of glutinous rice grain may leach out during the treatment. We did not evaluate the nutritional properties and rice composition after PEF treatment as we are interested in the physical attributes and microstructure properties of the PEF treated rice grains. Thus, we did not report the nutritional properties after the PEF-pulses treatments in this study.

  1. SEM analysis. Figure 1 does not report the different letters, nor image resolution: that is the dimension the bars are referred to. In particular, it is not clear if Figure 1a is referred to the control sample after 2h soaking or before, otherwise line 242 not shown has to be added. a. Figure 2 caption: angle and not angel!

Response 3: We have revised all figures that are included in the manuscript based on your suggestion. The caption of Fig. 1 was also revised to “SEM images of pre-soaked glutinous rice grain surface without PEF treatment (Control, a), and with 50 (b), 100 (c), 200 (d), and 300 (e) pulses of PEF treatment. Arrows indicate the pores or cavity on PEF treated rice grain surface. The SEM images captured for all samples provided a ×500 magnification and used scale bar of 50 μm.”

Figure 2 has been corrected, thank you for the catching the error (angle not angel).

  •  
  1. NMR analysis. Figure 4 is not well distinguishable. Both the ppm values and the spectra are not quite visible.

Which the procedure adopted by the authors to assign the resonances? In particular at Line 461 they refer to a literature study, but this is not shown in the Reference section [38]. Was it at the same field, temperature, experimental condition?

Line 470: higher resonance is as a matter of facts low field resonance and/or higher frequency.

Response 4: Figure 4 has been improved by redoing the labeling and scales. The reference citation and associated explanation have been added (see below).

According to Cheetham, & Tao, (1998), the native corn starch samples, with the apparent amylose content varied from 0% to 84%, were measured using the solid state 13C CP/MAS NMR at the magic-angle spinning rate of 2.5 KHz and decoupling field of 61 KHz. Spectra are referenced to external Me4Si via the low field resonance of adamantane (38.6 ppm) at room temperature. The spectra of corn starch (following figure) agree with the rice flour spectra presented in our study (Fig. 4).

The following contents have been added in this section, “According to Cheetham, & Tao (38), the shift of resonance peak and shoulder is due to the low field resonance of non-crystalline material. In our current study,”

Fig. 4.

  1. A Section reporting the Limitation of the study should be recommended.

Response 5: the following sentences have been added to the Conclusion Section, “Commercial glutinous rice products, such as rice cake, are composed of glutinous rice with other ingredients including sugar, oil and stabilizers, therefore studying their interactions during and after PEF treatment would be required to assess industrial applications, which were not investigated in this study. Thus, the gelling behaviors, gel textural properties and retrogradation behaviors of glutinous rice products prepared with PEF treatment warrant further research.”

Round 2

Reviewer 3 Report

In the light the changes made to the previous version of the manuscript, this reviewer thinks the paper now suitable for publishing in Foods journal.

Author Response

Thanks for your reviewed.